# Prevalence and Factors Associated with Long COVID Symptoms among U.S. Adults, 2022

**DOI:** 10.3390/vaccines12010099

**Published:** 2024-01-18

**Authors:** Kimberly H. Nguyen, Yingjun Bao, Julie Mortazavi, Jennifer D. Allen, Patricia O. Chocano-Bedoya, Laura Corlin

**Affiliations:** 1Hubert Department of Global Health, Emory University Rollins School of Public Health, Atlanta, GA 30322, USA; 2Department of Epidemiology, George Washington University Milken School of Public Health, Washington, DC 20037, USA; 3Department of Public Health & Community Medicine, Tufts University School of Medicine, Boston, MA 02111, USA; yingjun.bao@tufts.edu (Y.B.); julie-anne.mortazavi@tufts.edu (J.M.); laura.corlin@tufts.edu (L.C.); 4Department of Community Health, Tufts School of Arts and Sciences, Medford, MA 02115, USA; jennifer.allen@tufts.edu; 5Institute of Primary Health Care (BIHAM), University of Bern, 3012 Bern, Switzerland; patricia.chocano@unibe.ch; 6Department of Civil and Environmental Engineering, Tufts University School of Engineering, Medford, MA 02155, USA

**Keywords:** COVID-19 vaccination, vaccine hesitancy, vaccine confidence, COVID-19 outcomes, long COVID, disparities, symptoms, adults

## Abstract

Long COVID and its symptoms have not been examined in different subpopulations of U.S. adults. Using the 2022 BRFSS (n = 445,132), we assessed long COVID and each symptom by sociodemographic characteristics and health-related variables. Multivariable logistic regression was conducted to examine factors associated with long COVID and the individual symptoms. Prevalence differences were conducted to examine differences in long COVID by vaccination status. Overall, more than one in five adults who ever had COVID-19 reported symptoms consistent with long COVID (21.8%). The most common symptom was tiredness or fatigue (26.2%), followed by difficulty breathing or shortness of breath (18.9%), and loss of taste or smell (17.0%). Long COVID was more common among adults under 65 years, women, American Indian or Alaska Native or other/multi race group, smokers, and people with a disability, depression, overweight or obesity compared to their respective counterparts. The prevalence of long COVID was higher among unvaccinated adults (25.6%) than vaccinated adults (21.6%) overall, and for 20 of 32 subgroups assessed. These findings underscore the benefits of vaccination, the importance of early treatment, and the need to better inform health care resource allocation and support services for those experiencing long COVID.

## 1. Introduction

Despite the ending of the Public Health Emergency for COVID-19 in May 2023, lingering effects of COVID-19 remain [1,2]. While there have been recent increases in emergency department visits and hospitalizations in December 2023 compared to the previous month, little was known about the prevalence of and risk factors for long COVID among different subpopulations of adults in the U.S. [3] Long COVID is defined as signs, symptoms, and conditions that continue or develop after acute COVID-19 infection, and can last weeks, months, or years [4,5]. Symptoms of long COVID may include tiredness or fatigue that interferes with daily life, respiratory and heart symptoms, neurological symptoms, digestive symptoms, or other symptoms such as joint or muscle pain or rash [4]. Long COVID can have major impacts on mental health, quality of life, and the ability to work [6,7,8,9].

Studies suggest long COVID was more common among people who have experienced more severe COVID-19 illness, have underlying health conditions prior to contracting COVID-19, or did not receive a COVID-19 vaccine [4]. While the underlying causes are not well understood, early evidence suggests the risk of developing long COVID may be higher among some populations, such as people from racial or ethnic minority groups, people with disabilities, and people with lower levels of income or lack of health insurance [10]. Yet, much remains unknown. For example, it is unknown whether depression, smoking status, body mass index (BMI), and other factors affect the risk for long COVID, and if so, which symptoms are most prevalent. Previous studies have found these groups were more vulnerable to severe health outcomes and were less likely to be vaccinated against COVID-19 [11,12,13,14]. Being vaccinated against COVID-19 is one of the best ways to prevent long COVID [15,16]. In addition, being tested for and seeking early treatment for COVID-19 can minimize the potential for severe health consequences. Identifying the most frequent symptoms, as well as risk and protective factors, can help us to understand how we can best treat or support those with long COVID. Additionally, understanding which groups might be at greatest risk can help guide the development of targeted interventions, including vaccination programs, to those who may experience these long-term effects. 

The goal of this study was to examine factors associated with long COVID and its symptoms among a nationally representative sample of U.S. adults in 2022. The identification of groups with a higher prevalence of long COVID and the constellation of symptoms experienced across groups, overall and by vaccination status, could help increase efforts toward reducing disparities, treating or supporting patients, and preventing others from developing long-term effects through vaccination. 

## 2. Methods

### 2.1. Study Sample

The Behavioral Risk Factor Surveillance System (BRFSS) is an annual state-based system of telephone surveys conducted by the Centers for Disease Control and Prevention (CDC) that collected information on health risk behaviors, preventive health practices, and health care access primarily related to chronic disease and injury. In 2022, questions about long COVID were added to the core questionnaire that was asked in each state and U.S. territory, and a question on COVID-19 vaccination was included as an additional module in 27 states and jurisdictions (Arkansas, Connecticut, Delaware, Georgia, Guam, Hawaii, Idaho, Illinois, Iowa, Kansas, Louisiana, Montana, Nebraska, New Hampshire, New Jersey, New Mexico, North Carolina, North Dakota, Puerto Rico, Rhode Island, South Carolina, Tennessee, Texas, Virgin Islands, West Virginia, Wisconsin, and Wyoming). Relying on a disproportionate stratified sample design, the BRFSS uses a state-level, random digit dialed probability sample to select civilian noninstitutionalized adults aged 18 years and older residing in households. Interviews were conducted with one adult aged ≥18 years who was randomly selected to complete the survey within each household by landline or cellular telephone. Data were weighted to account for the probability of selection and adjust for non-response bias and noncoverage error. Raking (a weighting method) was conducted to increase representativeness of the BRFSS sample to the U.S. population based on age by sex, age groups by race and ethnicity, race and ethnicity groups, education levels, marital status, regions within states, gender by race and ethnicity, telephone source, and renter or owner status. The adult sample size and response rate was 445,132 and 45.1%, respectively [17]. Per the Emory University Institutional Review Board determination assessments, this study is not considered human subjects research.

### 2.2. COVID-19 Variables

COVID-19 diagnosis was defined as ever having been told by a doctor, nurse, or other health professional that [you] tested positive for COVID-19 or tested positive using a home test. Among those with a prior COVID-19 diagnosis, long COVID was defined as having any symptoms lasting 3 months or longer that were not experienced prior to having COVID-19. People who had a COVID-19 diagnosis were asked about the primary symptom they experienced: (1) tiredness or fatigue, (2) difficulty thinking or concentrating, (3) difficulty breathing or shortness of breath, (4) joint or muscle pain, (5) heart palpitations or chest pain, (6) dizziness on standing, (7) depression, anxiety, or mood changes, (8) worsening symptoms after physical or mental activities, (9) loss of taste or sense of smell, (10) other symptom, or (11) none. Due to the small sample sizes, responses for fast-beating or pounding heart or chest pain, dizziness, depression, anxiety, or mood changes, and symptoms worsening after physical or mental activities were combined with “other symptoms”. COVID-19 vaccination was defined as receiving at least one dose of a COVID-19 vaccination. 

### 2.3. Other Variables

Smoking status was categorized as current, former, or never smoker and defined as followed: “Current smoker” was someone who smoked at least 100 cigarettes and currently smoking some days or every day. “Former smoker” was someone who smoked at least 100 cigarettes and not currently smoking. “Never smoker” was someone who smoked 100 cigarettes or less. Depression was categorized as someone who has ever been told by a doctor or other health professional that [they] had a depressive disorder (including depression, major depression, dysthymia, or minor depression). Disability was characterized as someone who was deaf or have serious difficulty hearing; was blind or had serious difficulty seeing even when wearing glasses; had serious difficulty concentrating, remembering, or making decisions because of a physical, mental, or emotional condition; had difficulty dressing or bathing; had serious difficulty walking or climbing stairs; or had difficulty doing errands alone because of a physical, mental, or emotional condition. 

Sociodemographic variables assessed were age, sex, race/ethnicity, highest educational attainment, annual household income, Census region (based on state) [18] and body mass index (BMI; based on respondent’s self-reported weight and height).

### 2.4. Statistical Analysis

Sociodemographic characteristics were assessed for all adults overall and for adults with a prior diagnosis of COVID-19. The proportion of adults experiencing COVID-19 outcomes (long COVID and each primary symptom) was assessed overall and by each sociodemographic characteristic. The difference in proportion of adults experiencing each outcome was compared across levels of each variable (e.g., the proportion with long COVID among adults aged 18–49 years compared to those 65 years and older). Separate multivariable logistic regression models were conducted to examine (1) factors associated with long COVID and (2) factors associated with prior diagnosis of COVID-19, adjusted for age, sex, race/ethnicity, education level, income, health insurance, geographic region, BMI, smoking status, disability, and depression. Among a subset of the sample (27 states and jurisdictions that included COVID-19 vaccination questions in its survey), prevalence differences were conducted to examine differences in long COVID among vaccinated and non-vaccinated adults within levels of each sociodemographic characteristic and other health-related variables. All tests were two-tailed with the significance level set at α < 0.05. All analyses in this manuscript were calculated using Stata v18 to account for complex sample survey design and to produce nationally representative estimates. All results presented in the text of this manuscript were statistically significant.

## 3. Results

In the study population, 52.7% were 18–49 years, 58.3% were non-Hispanic (NH) White, 39.3% had a high school education or less, and 33.1% had an annual household income of ≥75,000 (Table 1). In addition, 67.5% had overweight or obesity, 63.0% were never smokers, 30.0% had a disability, 20.6% ever had depression, and 34.4% had a prior COVID-19 diagnosis. Among those with a prior COVID-19 diagnosis, 60.7% were between 18–49 years, 53.5% were female, 32.3% were college graduates or higher, 38.9% had annual household incomes of ≥$75,000, and 93.1% had health insurance. In addition, 70.4% had overweight or obesity, and 23.6% had depression.

Overall, more than one in five adults had long COVID (21.8%) among those who ever had a prior diagnosis of COVID-19 (Table 2). Adults who were 18–49 years or 50–59 years (adjusted odds ratio [aOR] = 1.38), female (aOR = 1.65), NH American Indian or Alaska Native (aOR = 1.29), or another/multiracial NH group (aOR = 1.30) were more likely to have had long COVID than their respective counterparts, while NH Asian (aOR = 0.58) and NH Black adults (aOR = 0.85) were less likely to have had long COVID compared to NH White adults. In addition, those with incomes < $75,000 compared to those with incomes ≥ $75,000 were more likely to have had long COVID, and those with health insurance were less likely (aOR = 0.82) to have had long COVID compared to those without health insurance. Those who lived in the Midwest (aOR = 1.15) or South (aOR = 1.16) were more likely to have had long COVID than those who lived in the Northeast. Moreover, those who had overweight or obesity (aOR = 1.31), were current (aOR = 1.13) or former smokers (aOR = 1.21), or had a disability (aOR = 1.58) or depression (aOR = 1.44) were more likely to have had long COVID compared to their respective counterparts.

The most common long COVID symptom was tiredness or fatigue (26.2%), followed by difficulty breathing or shortness of breath (18.9%), loss of taste or smell (17.0%), difficulty thinking or concentrating or forgetfulness/memory problems (9.8%), joint or muscle pain (6.2%), or other symptoms (16.7%) (Table 3). Younger adults (18–49 years) were less likely to have tiredness or fatigue (aOR = 0.53) but more likely to have loss of taste or smell (aOR = 1.66) or difficulty thinking or concentrating (aOR = 1.43) compared to older adults (≥65 years) (Table 4). Females were less likely to have difficulty breathing or shortness of breath (aOR = 0.76), and more likely to have difficulty thinking or concentrating (aOR = 1.30) or some other symptom (aOR = 1.18) compared to males. There were also differences in long COVID symptoms by race/ethnicity. NH Asian adults were less likely to have loss of taste or smell (aOR = 0.46) or difficulty thinking or concentrating (aOR = 0.40) compared to NH White adults. NH Black adults were less likely to have loss of taste or smell (aOR = 0.66) but more likely to have difficulty thinking or concentrating (aOR = 1.43) or joint or muscle pain (aOR = 1.85) compared to NH White adults. Hispanic adults were less likely to have loss of taste or smell (aOR = 0.78) but more likely to have joint or muscle pain (aOR = 2.10) compared to NH White adults.

Long COVID symptoms also varied among those with health-related conditions. Those with overweight or obesity were more likely to have difficulty breathing or shortness of breath (aOR = 1.27) but less likely to have loss of taste or smell (aOR = 0.76) compared to those without overweight or obesity (Table 4). Current smokers were more likely to have joint or muscle pain (aOR = 1.52) compared to never smokers. Those with a disability were more likely to have difficulty breathing or shortness of breath (aOR = 1.22) and difficulty thinking or concentrating (aOR = 1.36), and less likely to have loss of taste or smell (aOR = 0.80) compared to those without a disability. Those with depression were more likely to have difficulty thinking or concentrating (aOR = 1.25), and less likely to have loss of taste or smell (aOR = 0.80) compared to those without depression. 

Overall, the prevalence of long COVID differed by vaccination status. The prevalence of long COVID was 21.6% among vaccinated adults and 25.6% among non-vaccinated adults, for a difference of 4.0 percentage points (Table 5). In addition, long COVID was lower among vaccinated adults compared to non-vaccinated adults for 20 out of 32 subcategories, including all age groups and sexes. It was also lower among vaccinated adults compared to non-vaccinated adults for those who had overweight or obesity (prevalence difference [PD] = 4.2), former smokers (PD = 5.3), and those with depression (PD = 4.7).

## 4. Discussion

This study found long COVID was most common among younger age groups, females, NH American Indian or Alaska Native and NH other or multiracial groups, those with lower annual household income, those with no health insurance, and those living in the Midwest or South, which was consistent with other studies [19,20]. Long COVID was also more common among adults who had overweight or obesity, were current or former smokers, had a disability, or had depression. These represented vulnerable groups that were at greater risk for long COVID and at the same time, less likely to have been vaccinated against COVID-19, which could protect them from severe health consequences [21,22].

There were also differences by sociodemographic characteristics in common symptoms such as tiredness or fatigue, difficulty breathing or shortness of breath, loss of taste and smell, difficulty thinking or concentrating, or joint or muscle pain. Previous studies have found sex differences in long COVID, which could be related to differences in immune system function or sex hormones [23,24]. Other studies found long COVID was more likely with increasing age and BMI, which may also be related to immune function [25,26]. While our study has found higher prevalence of long COVID among younger age groups which was consistent with results from other nationally representative surveys [20], multiple factors could result in differences found in these studies, such as reporting of long COVID (self-report vs. doctor diagnosis), sample population (nationally representative data vs. medical records), and other sources of error (bias or misclassification). These results indicate differences in each long COVID symptom, particularly among different subpopulations, are not well understood. It is hypothesized these differences can stem from differences in exposure, frequency of infection, and severity of disease—each of which can have differential effects depending on factors such as age, medical conditions, and immune function [27,28]. Further studies are needed to elucidate the pathways through which these factors can impact the risk of long COVID. 

Not surprisingly, long COVID was less common among adults that had been vaccinated compared to adults that were not vaccinated. This finding supports existing evidence about the importance of vaccination in preventing long-term COVID-19 outcomes [15,16]. In addition, long COVID was also less common among vaccinated groups than unvaccinated groups for 20 out of 32 subgroups examined, such as adults who had overweight or obesity and those with depression, suggesting vaccination could help advance health equity and protect those who were most vulnerable to long-term health consequences of COVID-19. These results not only support previous literature [29], but also provide novel findings regarding the role of vaccination and long COVID among different subpopulations.

While the COVID-19 Public Health Emergency has ended, much work would be needed to reduce long-term harms of COVID-19 infections. Early COVID-19 testing and treatment can help prevent severe or long-term COVID-19 symptoms. Some programs have been found to increase access to COVID-19 testing in communities that were at greater risk of being affected by COVID-19, such as those with high transmission rates. For example, the CDC’s Increasing Community Access to Testing (ICATT) program provides no-cost COVID-19 testing for people without health insurance that have symptoms related to COVID-19 or who are exposed to someone with COVID-19 [30].

Before discussing implications of study findings, we acknowledge its limitations. First, the BRFSS is a cross-sectional study, so causal conclusions cannot be drawn. Second, COVID-19 vaccination, high-risk conditions, COVID-19 diagnosis, and long COVID symptoms were based on self-reports and were not verified by medical records, so may be subject to recall or social-desirability bias. Furthermore, the survey did not collect information on duration of symptoms, time since COVID-19 illness, or treatment during acute COVID infection, each of which could influence the reported prevalence of long COVID. Other nationally representative cross-sectional studies have shown the reported prevalence of long COVID in 2022 was 6.9–7.5%, which was lower than those found in this study [31,32]. However, other studies demonstrate the prevalence of long COVID can range from 7.5 to 41.0%, depending on how long COVID is measured and reported [28]. Over- or under-reporting of these variables may lead to misclassification, resulting in bias of the estimates. Despite these limitations, these results provide invaluable data on experiences of those with long COVID and factors associated with long COVID, including among persons who might not have accessed care, and may complement studies based on administrative data. Third, the subanalyses using the COVID-19 vaccination questions were only included in 27 states and territories, so those results may not be nationally representative. Fourth, the questionnaire did not specify the brand of COVID-19 vaccines, or the date of administration of each vaccine, in order to determine whether they have been up-to-date with all eligible doses. As a result, the effect of vaccination in this study may be underestimated. Finally, the survey completion response rates were 45.1%; the survey weighting adjustments may not entirely control for differences in coverage and non-response. 

Nevertheless, findings from this study offer several insights for practice. First, findings can aid in the identification of priority groups for intervention efforts. To our knowledge, prior health communication campaigns have not emphasized the risk of long COVID among groups that may be particularly vulnerable. We also note the need to provide guidance about individual rights, protections, and support for those experiencing long-term adverse effects from COVID-19 [33]. Promoting awareness of clinical guidance from the Department of Health and Human Services regarding treatment of those at greatest risk for developing long COVID is an important part of a comprehensive approach. It has been well established early COVID-19 testing and treatment could help prevent severe or long-term COVID-19 symptoms.

## 5. Conclusions

This study adds to the literature on the prevalence of long-term adverse health effects of COVID-19 and provides new information about population subgroups that may be at greatest risk. These findings corroborate calls to enhance efforts to diagnosis, treat, and mitigate the potential for adverse COVID-19 outcomes. Additionally, findings suggest the need to better educate the public about the benefits of vaccination, the importance of early treatment, the potentially harmful consequences of COVID-19 infections, and characteristics of those at greatest risk to better inform health care resource allocation and other support services for those experiencing long COVID. 

## Figures and Tables

**Table 1 vaccines-12-00099-t001:** Distribution of sociodemographic and other characteristics among adults, overall and among adults with a prior diagnosis of COVID-19; Behavioral Risk Factor Surveillance System, United States, 2022.

	Total	Among Adults with a Prior Diagnosis of COVID-19
	Weighted Percent (95% CI)	Weighted Percent (95% CI)
**Overall (unweighted N)**	445,132	124,313
**Age group (years)**		
18–49	52.7 (52.4, 53.0)	60.7 (60.1, 61.2)
50–64	24.2 (24.0, 24.5)	23.6 (23.2, 24.1)
≥65	23.1 (22.9, 23.3)	15.7 (15.3, 16.1)
**Sex**		
Males	48.7 (48.4, 49.0)	46.5 (46.0, 47.1)
Females	51.3 (51.0, 51.6)	53.5 (52.9, 54.0)
**Race/ethnicity**		
NH White	58.3 (57.9, 58.6)	58.7 (58.1, 59.2)
NH American Indian or Alaska Native	1.2 (1.1, 1.3)	1.1 (1.0, 1.2)
NH Asian	6.2 (6.0, 6.4)	5.7 (5.3, 6.1)
NH Black or African American	11.9 (11.6, 12.1)	10.7 (10.4, 11.1)
Hispanic	18.8 (18.5, 19.1)	19.9 (19.4, 20.5)
NH Other	3.7 (3.5, 3.8)	3.8 (3.6, 4.1)
**Education level**		
High school graduate or less	39.3 (38.9, 39.6)	35.5 (34.9, 36.1)
Some college or technical school	30.2 (29.9, 30.5)	32.2 (31.7, 32.8)
College graduate or more	30.5 (30.3, 30.8)	32.3 (31.8, 32.8)
**Income**		
<$25,000	13.5 (13.3, 13.7)	11.3 (10.9, 11.7)
$25,000 to <$50,000	20.1 (19.9, 20.4)	19.7 (19.2, 20.1)
$50,000 to <$75,000	12.4 (12.2, 12.6)	13.3 (12.9, 13.7)
≥$75,000	33.1 (32.8, 33.4)	38.9 (38.4, 39.5)
Don’t know/refused	20.8 (20.5, 21.1)	16.8 (16.4, 17.3)
**Health Insurance**		
No	8.5 (8.3, 8.7)	6.9 (6.6, 7.2)
Yes	91.5 (91.3, 91.7)	93.1 (92.8, 93.4)
**Region**		
Northeast	17.5 (17.3, 17.6)	17.5 (17.2, 17.8)
Midwest	20.6 (20.4, 20.7)	20.6 (20.4, 20.9)
South	38.3 (38.1, 38.5)	38.4 (38.0, 38.7)
West	23.7 (23.5, 23.9)	23.5 (23.1, 23.9)
**BMI**		
Normal weight/underweight	32.5 (32.2, 32.8)	29.6 (29.1, 30.2)
Overweight and obese	67.5 (67.2, 67.8)	70.4 (69.8, 70.9)
**Smoking status**		
Never smoker	63.0 (62.7, 63.3)	66.0 (65.4, 66.5)
Current smoker	12.8 (12.6, 13.1)	10.1 (9.8, 10.4)
Former smoker	24.1 (23.9, 24.4)	24.0 (23.5, 24.4)
**Disability**		
Without disability	70.0 (69.7, 70.3)	71.9 (71.4, 72.4)
With disability	30.0 (29.7, 30.3)	28.1 (27.6, 28.6)
**Depression**		
No	79.4 (79.1, 79.6)	76.4 (76.0, 76.9)
Yes	20.6 (20.4, 20.9)	23.6 (23.1, 24.0)
**COVID-19 vaccination (≥1 dose) ^a^**		
No	21.2 (20.8, 21.7)	23.4 (22.6, 24.2)
Yes	78.8 (78.3, 79.2)	76.6 (75.8, 77.4)
**Prior COVID-19 Diagnosis**		
Yes	34.4 (34.1, 34.7)	-
No	65.6 (65.3, 65.9)	-

Note: All values are weighted except as noted. Abbreviations: CI = confidence interval; NH = non-Hispanic; BMI = body mass index; ^a^ COVID-19 vaccination was added as an additional module in only 27 states and jurisdictions.

**Table 2 vaccines-12-00099-t002:** Prevalence of and factors associated with COVID-19 outcomes, Behavioral Risk Factor Surveillance System, United States, 2022.

	Prior Diagnosis of COVID-19		Long COVID	
	% (95% CI)	aOR (95% CI)	% (95%CI)	aOR (95% CI) ^a^
**Overall**	34.4 (34.1, 34.7)		21.8 (21.4, 22.3)	
**Age group (years)**				
18–49	40.2 (39.7, 40.7) *	2.32 (2.23, 2.43)	22.1 (21.5, 22.7) *	1.38 (1.26, 1.51)
50–64	33.6 (33.0, 34.2) *	1.65 (1.58, 1.73)	23.3 (22.4, 24.3) *	1.38 (1.26, 1.53)
≥65 (Reference)	23.1 (22.5, 23.6)	1.0	19.1 (18.1, 20.1)	1.0
**Sex**				
Males (Reference)	32.8 (32.3, 33.2)	1.0	17.2 (16.6, 17.9)	1.0
Females	36.0 (35.5, 36.4) *	1.20 (1.17, 1.24)	25.9 (25.2, 26.6) *	1.65 (1.55, 1.76)
**Race/ethnicity**				
NH White (Reference)	34.0 (33.7, 34.4)	1.0	22.2 (21.7, 22.8)	1.0
NH American Indian or Alaska Native	33.4 (30.8, 36.0)	1.01 (0.88, 1.15)	29.1 (25.1, 33.2) *	1.29 (1.04, 1.61)
NH Asian	33.1 (31.3, 34.9)	0.88 (0.80, 0.97)	11.8 (9.5, 14.2) *	0.58 (0.45, 0.75)
NH Black or African American	32.6 (31.6, 33.5) *	0.91 (0.87, 0.96)	20.2 (18.8, 21.7) *	0.85 (0.76, 0.95)
Hispanic	37.7 (36.7, 38.7) *	1.18 (1.11, 1.24)	22.6 (21.2, 24.0)	1.01 (0.91, 1.11)
NH Other	35.7 (33.8, 37.7)	0.99 (0.91, 1.09)	27.5 (24.5, 30.6) *	1.30 (1.10, 1.54)
**Education level**				
High school graduate or less	31.7 (31.1, 32.2) *	0.98 (0.94, 1.02)	22.2 (21.4, 23.1) *	1.04 (0.96, 1.13)
Some college or technical school	36.4 (35.8, 37.0)	1.10 (1.06, 1.14)	24.7 (23.8, 25.6) *	1.23 (1.14, 1.32)
College graduate or more (Reference)	35.9 (35.5, 36.4)	1.0	18.5 (17.9, 19.2)	1.0
**Income**				
<$25,000	28.5 (27.6, 29.4) *	0.66 (0.62, 0.70)	27.4 (25.8, 29.1) *	1.18 (1.05, 1.33)
$25,000 to <$50,000	33.4 (32.7, 34.1) *	0.82 (0.78, 0.86)	24.2 (23.2, 25.3) *	1.10 (1.01, 1.20)
$50,000 to <$75,000	36.2 (35.3, 37.1) *	0.91 (0.87, 0.96)	23.5 (22.2, 24.8) *	1.14 (1.04, 1.25)
≥$75,000 (Reference)	39.6 (39.1, 40.1)	1.0	19.3 (18.6, 20.0)	1.0
Don’t know/refused	29.4 (28.7, 30.2) *	0.72 (0.69, 0.76)	19.8 (18.6, 20.9)	0.95 (0.86, 1.06)
**Health Insurance**				
No	29.0 (27.7, 30.2)	1.0	26.1 (24.0, 28.3)	1.0
Yes	35.1 (34.8, 35.4) *	1.42 (1.32, 1.52)	21.7 (21.2, 22.2) *	0.82 (0.71, 0.93)
**Region**				
Northeast (Reference)	34.5 (33.8, 35.2)	1.0	19.6 (18.5, 20.7)	1.0
Midwest	34.1 (33.6, 34.6)	0.97 (0.92, 1.01)	23.0 (22.2, 23.8) *	1.15 (1.05, 1.26)
South	34.5 (34.0, 35.1)	1.01 (0.96, 1.05)	23.0 (22.2, 23.8) *	1.16 (1.06, 1.27)
West	34.2 (33.5, 34.9)	0.95 (0.90, 1.00)	20.9 (19.8, 22.0)	1.06 (0.95, 1.17)
**BMI**				
Normal weight/underweight (Reference)	31.8 (31.3, 32.4)	1.0	18.5 (17.6, 19.3)	1.0
Overweight and obese	36.0 (35.6, 36.4) *	1.22 (1.18, 1.27)	23.2 (22.7, 23.8) *	1.31 (1.22, 1.41)
**Smoking status**				
Never smoker (Reference)	35.9 (35.5, 36.3)	1.0	20.2 (19.6, 20.8)	1.0
Current smoker	27.6 (26.8, 28.4) *	0.70 (0.67, 0.74)	26.7 (25.1, 28.2) *	1.13 (1.02, 1.25)
Former smoker	34.3 (33.6, 34.9) *	1.05 (1.01, 1.09)	24.5 (23.5, 25.4) *	1.21 (1.12, 1.30)
**Disability**				
Without disability (Reference)	35.4 (35.0, 35.8)	1.0	18.5 (18.0, 19.1)	1.0
With disability	32.3 (31.7, 32.9) *	1.03 (0.99, 1.07)	30.3 (29.3, 31.2) *	1.58 (1.47, 1.70)
**Depression**				
No (Reference)	33.4 (33.0, 33.8)	1.0	18.9 (18.3, 19.4)	1.0
Yes	38.2 (37.5, 38.9) *	1.14 (1.10, 1.19)	31.4 (30.4, 32.5) *	1.44 (1.34, 1.55)
**COVID-19 vaccination (≥1 dose) ^b^**				
No (Reference)	38.3 (37.1, 39.5)	^c^	25.6 (23.9, 27.3)	^c^
Yes	33.9 (33.3, 34.4) *	^c^	21.6 (20.8, 22.5) *	^c^

Note: All values are weighted. Abbreviations: aOR = adjusted odds ratio; CI = confidence interval; NH = non-Hispanic; * *p* value < 0.05 in a *t*-test comparing the % of each category (e.g., Non-Hispanic Black) in a given criteria with the reference category (e.g., Non-Hispanic White) in that given criteria. ^a^ Adjusted for age, sex, race/ethnicity, education level, income, health insurance, region, BMI, smoking status, disability, and depression. N = 102,792. ^b^ COVID-19 vaccination was added as an additional module in only 27 states and jurisdictions. ^c^ Not included in multivariable model.

**Table 3 vaccines-12-00099-t003:** Prevalence of long COVID symptoms by sociodemographic factors, Behavioral Risk Factor Surveillance System, United States, 2022.

	Tiredness or Fatigue	Difficulty Breathing or Shortness of Breath	Loss of Taste or Smell	Difficulty Thinking or Concentrating	Joint or Muscle Pain	Some Other Symptom
	% (95%CI)	% (95%CI)	% (95%CI)	% (95%CI)	% (95%CI)	% (95%CI)
Unweighted n with symptom	7072	4772	4234	2564	1378	4209
**Overall**	26.2 (25.1, 27.2)	18.9 (17.9, 19.8)	17.0 (16.1, 18.0)	9.8 (9.0, 10.5)	6.2 (5.6, 6.9)	16.7 (15.7, 17.7)
**Age group (years)**						
18–49	23.2 (21.9, 24.6) *	18.6 (17.3, 19.9)	18.7 (17.4, 19.9) *	9.7 (8.7, 10.7) *	6.1 (5.2, 7.0)	17.9 (16.5, 19.4)
50–64	28.8 (26.7, 30.8) *	19.2 (17.4, 20.9)	15.2 (13.2, 17.2)	11.5 (10.0, 13.0) *	6.7 (5.5, 7.9)	14.5 (13.1, 16.0)
≥65 (Reference)	34.4 (31.4, 37.3)	19.7 (17.2, 22.2)	13.6 (11.5, 15.7)	6.9 (5.7, 8.1)	5.4 (3.8, 7.0)	15.3 (13.0, 17.6)
**Sex**						
Males (Reference)	24.9 (23.2, 26.6)	20.8 (19.1, 22.4)	18.8 (17.1, 20.5)	8.2 (7.2, 9.2)	6.0 (5.0, 7.1)	15.4 (14.0, 16.8)
Females	26.9 (25.5, 28.2)	17.8 (16.6, 19.0) *	16.0 (14.9, 17.1) *	10.7 (9.7, 11.8) *	6.4 (5.5, 7.2)	17.5 (16.1, 18.8) *
**Race/ethnicity**						
NH White (Reference)	27.0 (25.8, 28.2)	18.4 (17.4, 19.5)	18.5 (17.5, 19.6)	9.7 (9.0, 10.4)	4.5 (3.8, 5.1)	16.1 (15.2, 17.1)
NH American Indian or Alaska Native	28.7 (21.5, 35.9)	21.3 (14.0, 28.6)	15.9 (10.7, 21.0)	8.2 (5.2, 11.1)	3.5 (1.8, 5.3)	20.8 (11.9, 29.7)
NH Asian	25.0 (15.9, 34.1)	17.5 (9.2, 25.8)	11.6 (5.8, 17.5) *	5.7 (2.4, 9.0) *	6.1 (1.3, 10.9)	30.2 (19.4, 41.0) *
NH Black or African American	25.6 (22.1, 29.1)	21.4 (18.2, 24.6)	12.6 (9.9, 15.3) *	12.5 (9.5, 15.5)	8.6 (5.8, 11.3) *	15.5 (12.7, 18.3)
Hispanic	24.9 (22.0, 27.8)	18.1 (15.6, 20.6)	15.4 (12.6, 18.3) *	9.0 (6.6, 11.3)	10.3 (8.3, 12.4) *	17.5 (14.4, 20.7)
NH Other	22.4 (16.7, 28.1)	22.4 (16.4, 28.3)	18.6 (14.0, 23.3)	10.5 (5.5, 15.5)	5.6 (3.1, 8.1)	14.5 (10.7, 18.3)
**Education level**						
High school graduate or less	24.6 (22.7, 26.4) *	20.8 (18.9, 22.7) *	19.0 (17.1, 20.8) *	7.5 (6.4, 8.5) *	7.1 (5.8, 8.3)	15.4 (13.6, 17.3) *
Some college or technical school	27.0 (25.1, 28.8)	19.3 (17.7, 20.9) *	15.9 (14.5, 17.4)	9.7 (8.2, 11.2) *	5.9 (4.8, 7.1)	17.2 (15.5, 18.9)
College graduate or more (Reference)	27.3 (25.6, 29.1)	15.7 (14.3, 17.0)	15.8 (14.4, 17.1)	13.0 (11.6, 14.4)	5.5 (4.6, 6.5)	17.8 (16.4, 19.3)
**Income**						
<$25,000	27.4 (24.4, 30.5)	21.0 (18.2, 23.8) *	14.1 (11.7, 16.6) *	7.1 (5.6, 8.5) *	7.1 (5.2, 9.1) *	18.6 (15.1, 22.1)
$25,000 to <$50,000	24.2 (22.1, 26.2)	22.5 (20.2, 24.8) *	18.1 (15.8, 20.4)	7.6 (6.5, 8.8) *	7.8 (6.1, 9.5) *	15.3 (13.6, 16.9) *
$50,000 to <$75,000	27.5 (24.8, 30.3)	17.9 (15.6, 20.1)	17.0 (14.7, 19.3)	11.6 (9.2, 14.0)	5.5 (3.8, 7.3)	15.1 (13.0, 17.2)
≥$75,000 (Reference)	25.9 (24.1, 27.8)	16.1 (14.6, 17.5)	17.9 (16.4, 19.5)	12.4 (10.8, 13.9)	4.5 (3.7, 5.2)	17.7 (15.9, 19.5)
Don’t know/refused	27.1 (24.4, 29.8)	18.9 (16.3, 21.5)	16.2 (13.8, 18.6)	7.9 (6.5, 9.3) *	7.9 (5.7, 10.0) *	16.2 (13.9, 18.5)
**Health Insurance**						
No (Reference)	24.7 (21.0, 28.5)	21.7 (17.4, 26.1)	18.0 (14.0, 21.9)	7.6 (5.2, 10.0)	9.2 (6.1, 12.2)	15.1 (11.2, 18.9)
Yes	26.5 (25.4, 27.7)	18.5 (17.5, 19.5)	16.9 (15.9, 17.9)	10.1 (9.3, 11.0) *	5.9 (5.2, 6.6) *	16.9 (15.8, 17.9)
**Region**						
Northeast (Reference)	27.7 (24.9, 30.5)	20.6 (17.8, 23.3)	15.2 (13.1, 17.4)	9.0 (7.4, 10.5)	6.5 (5.0, 8.0)	16.0 (14.1, 18.0)
Midwest	24.4 (22.7, 26.0) *	18.2 (16.6, 19.7)	20.4 (18.7, 22.0) *	9.3 (8.1, 10.5)	4.6 (3.8, 5.4) *	17.4 (16.0, 18.9)
South	26.8 (25.0, 28.6)	19.1 (17.5, 20.7)	16.2 (14.6, 17.7)	9.8 (8.6, 10.9)	7.2 (5.8, 8.5)	15.2 (13.7, 16.6)
West	25.1 (22.6, 27.6)	17.9 (15.8, 20.0)	17.5 (15.2, 19.9)	11.2 (9.0, 13.4)	5.4 (4.2, 6.6)	19.7 (16.7, 22.7) *
**BMI**						
Normal weight/underweight (Reference)	24.1 (22.1, 26.1)	16.1 (14.1, 18.0)	20.3 (18.2, 22.4)	9.2 (7.6, 10.8)	5.6 (4.5, 6.8)	17.9 (16.1, 19.7)
Overweight and obese	26.9 (25.6, 28.2) *	19.8 (18.6, 20.9) *	16.0 (15.0, 17.0) *	10.0 (9.0, 10.9)	6.2 (5.4, 7.0)	16.4 (15.1, 17.6)
**Smoking status**						
Never smoker (Reference)	25.7 (24.4, 27.1)	18.1 (16.9, 19.4)	17.1 (15.9, 18.4)	9.7 (8.7, 10.8)	6.4 (5.5, 7.2)	17.2 (15.9, 18.5)
Current smoker	26.4 (23.4, 29.5)	20.3 (17.1, 23.4)	16.9 (14.6, 19.3)	8.7 (7.1, 10.3)	8.2 (5.9, 10.6)	14.8 (12.5, 17.2)
Former smoker	26.9 (24.9, 28.8)	19.9 (18.1, 21.6)	16.8 (15.1, 18.5)	10.5 (9.1, 11.8)	5.0 (3.8, 6.2)	16.5 (14.6, 18.3)
**Disability**						
Without disability (Reference)	25.6 (24.2, 27.0)	17.1 (15.9, 18.3)	18.9 (17.6, 20.2)	9.1 (8.1, 10.1)	6.5 (5.5, 7.4)	16.8 (15.5, 18.1)
With disability	27.1 (25.5, 28.8)	21.6 (20.0, 23.2) *	14.2 (12.9, 15.6) *	10.8 (9.6, 12.1) *	5.7 (4.8, 6.6)	16.5 (14.9, 18.1)
**Depression**						
No (Reference)	25.9 (24.6, 27.2)	18.2 (17.1, 19.4)	17.9 (16.7, 19.1)	8.8 (7.9, 9.8)	6.9 (6.0, 7.7)	16.3 (15.1, 17.5)
Yes	26.7 (24.9, 28.4)	20.2 (18.4, 22.0)	14.9 (13.5, 16.3) *	11.7 (10.4, 13.0) *	5.1 (4.1, 6.1) *	17.7 (16.0, 19.5)

Note: All values are weighted except as noted. Abbreviations: CI = confidence interval; NH = non-Hispanic; BMI = body mass index; * *p* value < 0.05 in a *t*-test comparing the % of each category (e.g., Non-Hispanic Black) in a given criteria with the reference category (e.g., Non-Hispanic White) in that given criteria.

**Table 4 vaccines-12-00099-t004:** Factors associated with long COVID symptoms, Behavioral Risk Factor Surveillance System, United States, 2022.

	Tiredness or Fatigue	Difficulty Breathing or Shortness of Breath	Loss of Taste or Smell	Difficulty Thinking or Concentrating	Joint or Muscle Pain	Some Other Symptom
	aOR (95% CI) ^a^	aOR (95% CI)	aOR (95% CI)	aOR (95% CI)	aOR (95% CI)	aOR (95% CI)
**Age group (years)**						
18–49	0.53 (0.44, 0.63)	0.98 (0.80, 1.19)	1.66 (1.36, 2.02)	1.43 (1.12, 1.83)	0.94 (0.63, 1.40)	1.26 (0.99, 1.62)
50–64	0.75 (0.62, 0.90)	1.01 (0.82, 1.25)	1.18 (0.96, 1.46)	1.59 (1.21, 2.08)	1.23 (0.82, 1.83)	0.95 (0.75, 1.22)
≥65 (Reference)	1.0	1.0	1.0	1.0	1.0	1.0
**Sex**						
Males (Reference)	1.0	1.0	1.0	1.0	1.0	1.0
Females	1.11 (0.98, 1.26)	0.76 (0.66, 0.87)	0.86 (0.75, 1.00)	1.30 (1.07, 1.58)	1.05 (0.81, 1.35)	1.18 (1.01, 1.39)
**Race/ethnicity**						
NH White (Reference)	1.0	1.0	1.0	1.0	1.0	1.0
NH American Indian or Alaska Native	1.20 (0.83, 1.75)	1.01 (0.69, 1.48)	0.86 (0.57, 1.31)	0.90 (0.58, 1.40)	0.82 (0.45, 1.49)	1.33 (0.72, 2.43)
NH Asian	1.15 (0.69, 1.92)	1.17 (0.66, 2.07)	0.46 (0.23, 0.90)	0.40 (0.20, 0.80)	1.50 (0.59, 3.83)	1.96 (1.13, 3.42)
NH Black or African American	0.97 (0.78, 1.20)	1.23 (0.98, 1.53)	0.66 (0.50, 0.87)	1.43 (1.04, 1.96)	1.85 (1.23, 2.77)	0.92 (0.71, 1.18)
Hispanic	0.95 (0.78, 1.16)	0.92 (0.74, 1.14)	0.78 (0.62, 0.99)	1.08 (0.78, 1.48)	2.10 (1.52, 2.91)	1.17 (0.93, 1.48)
NH Other	0.76 (0.53, 1.08)	1.28 (0.87, 1.87)	1.05 (0.75, 1.46)	1.10 (0.64, 1.89)	1.29 (0.76, 2.17)	0.82 (0.58, 1.16)
**Education level**						
High school graduate or less	0.92 (0.79, 1.07)	1.19 (0.99, 1.44)	1.29 (1.08, 1.55)	0.63 (0.49, 0.82)	0.89 (0.66, 1.20)	0.89 (0.74, 1.08)
Some college or technical school	1.02 (0.87, 1.18)	1.16 (0.98, 1.37)	1.09 (0.92, 1.28)	0.74 (0.58, 0.94)	0.98 (0.72, 1.34)	0.94 (0.78, 1.13)
College graduate or more (Reference)	1.0	1.0	1.0	1.0	1.0	1.0
**Income**						
<$25,000	1.04 (0.83, 1.29)	1.27 (1.01, 1.59)	0.81 (0.62, 1.04)	0.53 (0.37, 0.75)	1.43 (0.96, 2.13)	1.16 (0.89, 1.53)
$25,000 to <$50,000	0.89 (0.75, 1.05)	1.45 (1.20, 1.76)	0.96 (0.79, 1.16)	0.62 (0.48, 0.82)	1.66 (1.19, 2.30)	0.90 (0.73, 1.10)
$50,000 to <$75,000	1.05 (0.87, 1.26)	1.05 (0.86, 1.30)	0.97 (0.79, 1.19)	0.96 (0.71, 1.29)	1.17 (0.79, 1.74)	0.89 (0.72, 1.12)
≥$75,000 (Reference)	1.0	1.0	1.0	1.0	1.0	1.0
Don’t know/refused	1.00 (0.82, 1.22)	1.17 (0.91, 1.50)	0.96 (0.75, 1.23)	0.63 (0.46, 0.86)	1.45 (1.00, 2.09)	1.01 (0.79, 1.29)
**Health Insurance**						
No (Reference)	1.0	1.0	1.0	1.0	1.0	1.0
Yes	0.91 (0.72, 1.16)	0.98 (0.73, 1.32)	1.01 (0.74, 1.38)	1.08 (0.74, 1.59)	0.79 (0.50, 1.23)	1.10 (0.79, 1.53)
**Region**						
Northeast (Reference)	1.0	1.0	1.0	1.0	1.0	1.0
Midwest	0.83 (0.69, 1.00)	0.88 (0.71, 1.09)	1.39 (1.12, 1.73)	1.03 (0.80, 1.34)	0.75 (0.54, 1.03)	1.12 (0.92, 1.37)
South	0.95 (0.79, 1.14)	0.86 (0.70, 1.07)	1.15 (0.92, 1.42)	1.08 (0.84, 1.39)	1.05 (0.77, 1.44)	0.98 (0.80, 1.20)
West	0.91 (0.74, 1.11)	0.90 (0.71, 1.14)	1.20 (0.95, 1.51)	1.22 (0.89, 1.67)	0.66 (0.47, 0.94)	1.24 (0.98, 1.59)
**BMI**						
Normal weight/underweight (Reference)	1.0	1.0	1.0	1.0	1.0	1.0
Overweight and obese	1.14 (0.99, 1.31)	1.27 (1.06, 1.51)	0.76 (0.65, 0.90)	1.06 (0.84, 1.34)	1.10 (0.84, 1.45)	0.91 (0.78, 1.07)
**Smoking status**						
Never smoker (Reference)	1.0	1.0	1.0	1.0	1.0	1.0
Current smoker	1.00 (0.83, 1.22)	0.99 (0.78, 1.25)	1.02 (0.83, 1.25)	0.96 (0.75, 1.22)	1.52 (1.03, 2.25)	0.91 (0.73, 1.15)
Former smoker	0.98 (0.85, 1.12)	1.05 (0.89, 1.23)	1.01 (0.87, 1.18)	1.08 (0.89, 1.31)	0.86 (0.64, 1.15)	1.05 (0.88, 1.26)
**Disability**						
Without disability (Reference)	1.0	1.0	1.0	1.0	1.0	1.0
With disability	1.02 (0.89, 1.18)	1.22 (1.05, 1.41)	0.80 (0.68, 0.94)	1.36 (1.11, 1.66)	0.77 (0.57, 1.04)	0.97 (0.81, 1.17)
**Depression**						
No (Reference)	1.0	1.0	1.0	1.0	1.0	1.0
Yes	1.11 (0.96, 1.27)	1.08 (0.92, 1.27)	0.80 (0.68, 0.95)	1.25 (1.02, 1.54)	0.76 (0.56, 1.02)	1.06 (0.87, 1.28)

Note: All values are weighted except as noted. Abbreviations: CI = confidence interval; aOR = adjusted odds ratio; NH = non-Hispanic; BMI = body mass index; ^a^ Adjusted for age, sex, race/ethnicity, education level, income, health insurance, region, BMI, smoking status, disability, and depression.

**Table 5 vaccines-12-00099-t005:** Prevalence and prevalence difference of long COVID by sociodemographic characteristics among vaccinated and unvaccinated adults, ^a^ Behavioral Risk Factor Surveillance System, United States, 2022.

	Vaccinated (≥1 Dose)	Not Vaccinated	Prevalence Difference
	% (95%CI)	% (95%CI)	% (95%CI)
**Overall**	21.6 (20.8, 22.5)	25.6 (23.9, 27.3)	4.0 (2.1, 5.8)
**Age group (years)**			
18–49	22.2 (21.0, 23.4)	25.3 (23.2, 27.4)	3.1 (0.6, 5.5)
50–64	22.5 (20.9, 24.0)	27.6 (24.3, 31.0)	5.2 (1.5, 8.9)
≥65	18.9 (17.4, 20.4)	24.9 (20.3, 29.5)	6.0 (1.1, 10.9)
**Sex**			
Males	17.0 (15.9, 18.1)	19.9 (17.7, 22.2)	3.0 (0.5, 5.5)
Females	25.5 (24.3, 26.7)	31.1 (28.6, 33.6)	5.6 (2.8, 8.3)
**Race/ethnicity**			
NH White	22.0 (21.0, 23.0)	26.2 (24.3, 28.0)	4.2 (2.1, 6.3)
NH American Indian or Alaska Native	29.7 (22.5, 36.9)	17.9 (8.9, 26.9)	−11.8 (−24.1, 0.6)
NH Asian	^b^		^b^
NH Black or African American	20.9 (18.2, 23.5)	22.8 (17.0, 28.7)	1.9 (−4.4, 8.3)
Hispanic	^b^	26.4 (20.7, 32.0)	^b^
NH Other	22.2 (17.4, 26.9)	29.5 (21.4, 37.7)	7.4 (−2.0, 16.8)
**Education level**			
High school graduate or less	21.7 (20.1, 23.3)	25.0 (22.4, 27.6)	3.3 (0.3, 6.3)
Some college or technical school	24.7 (23.0, 26.3)	27.3 (24.6, 30.0)	2.6 (−0.6, 5.8)
College graduate or more	18.8 (17.8, 19.9)	23.6 (20.4, 26.8)	4.8 (1.4, 8.2)
**Income**			
<$25,000	25.0 (22.5, 27.5)	32.6 (27.1, 38.1)	7.6 (1.6, 13.6)
$25,000 to <$50,000	24.5 (22.5, 26.5)	27.7 (24.3, 31.1)	3.2 (−0.7, 7.2)
$50,000 to <$75,000	22.6 (20.4, 24.7)	26.2 (22.2, 30.3)	3.7 (−0.9, 8.3)
≥$75,000	19.4 (18.2, 20.7)	23.1 (20.3, 25.9)	3.7 (0.6, 6.8)
Don’t know/refused	20.2 (18.1, 22.4)	20.7 (17.0, 24.4)	0.5 (−3.8, 4.8)
**Health Insurance**			
No	27.1 (22.7, 31.6)	29.7 (23.5, 36.0)	2.6 (−4.9, 10.1)
Yes	21.4 (20.5, 22.2)	25.3 (23.5, 27.0)	3.9 (2.0, 5.8)
**Region**			
Northeast	19.3 (17.6, 20.9)	22.8 (18.0, 27.7)	3.6 (−1.5, 8.7)
Midwest	21.1 (19.4, 22.7)	24.9 (21.7, 28.1)	3.9 (0.3, 7.5)
South	22.8 (21.5, 24.1)	26.2 (23.9, 28.6)	3.4 (0.7, 6.1)
West	22.6 (21.0, 24.2)	27.4 (24.8, 30.0)	4.8 (1.8, 7.9)
**BMI**			
Normal weight/underweight	18.2 (16.7, 19.8)	21.1 (18.2, 24.1)	2.9 (−0.5, 6.2)
Overweight and obese	23.0 (22.0, 24.0)	27.2 (25.1, 29.3)	4.2 (1.9, 6.5)
**Smoking status**			
Never smoker	20.2 (19.2, 21.2)	23.4 (21.1, 25.7)	3.2 (0.7, 5.7)
Current smoker	26.1 (22.7, 29.4)	28.0 (23.9, 32.1)	1.9 (−3.4, 7.2)
Former smoker	24.2 (22.5, 25.9)	29.6 (26.5, 32.7)	5.3 (1.8, 8.9)
**Disability**			
Without disability	18.6 (17.6, 19.5)	22.5 (20.5, 24.6)	4.0 (1.7, 6.2)
With disability	29.4 (27.7, 31.1)	32.8 (29.7, 35.9)	3.4 (−0.1, 6.9)
**Depression**			
No	18.7 (17.9, 19.6)	22.3 (20.4, 24.2)	3.5 (1.5, 5.6)
Yes	31.5 (29.4, 33.5)	36.2 (32.7, 39.7)	4.7 (0.7, 8.8)

Note: All values are weighted. Abbreviations: CI = confidence interval; NH = non-Hispanic; BMI = body mass index; ^a^ COVID-19 vaccination was added as an additional module in only 27 states and jurisdictions. ^b^ Cell suppressed when residual standard error was greater than 30%.

## Data Availability

The data that support the findings of this study are openly available at https://www.cdc.gov/brfss/data_documentation/index.htm.

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
