# Peer review of "Prevalence and Factors Associated with Long COVID Symptoms among U.S. Adults, 2022"

_vaccines, 2024, doi:10.3390/vaccines12010099_

Round 1

Reviewer 1 Report

Comments and Suggestions for Authors

The manuscript is potentially interesting, but has major flaws in explanation of applied methods, and in the presentation and discussion of study results.

Major issues

1)      Methods, Statistical Analysis: authors state that prevalence point estimates with 95% Confidence Intervals were obtained taking into account the complex sampling strategy. Presumably, authors accounted for sampling methods also when estimating prevalence differences (Table 5) or when applying logistic regression (text, Table 4): please specify. As regards logistic regression, authors present aOR (adjusted odds ratio): adjusted for what (only in a footnote of Table 4 adjustment variables are reported)?

2)      Results, Tables 2-5: please delete footnotes on smoking status, disability, depression (notes b-d in Table 2): the variables are extensively explained in the Methods, are notes are too lengthy

3)      Results, Table 1: the most important study variable is missing, e.g. the prevalence of subjects ever diagnosed with COVID-19. The latter represents the denominator for the prevalence of long-COVID, and MUST be absolutely reported and discussed (including possible reasons for underdiagnoses and underreporting, and possible bias introduced in the study)

4)      The same analyses reported in Table 1 should be therefore replicated for subjects reporting COVID-19 (see comment above)

5)      Table 4 might be deleted: aOR are already reported in the text, and all the relevant information is already displayed in Table 3

6)      Discussion is poor and must be more focused on study findings compared to previous evidence: as an example, authors cite studies reporting that long COVID is more likely with increasing age, which is the exact contrary of study results. Author might also cite a preprint based on BRFSS findings (doi 10.1101/2023.09.29.23296352).

Minor comments:

1)      In the second sentence of the introduction, authors cite an increase in hospitalization in December 2023, which was a too early finding at the time of the submission (the reference website was accessed in October 2023)

Comments on the Quality of English Language

The manuscript must be extensively edited, as an example, last sentence of the Introduction, “The identification OF groups…”, and other typos.

Author Response

The authors greatly appreciate the prompt and constructive comments received by the reviewers. The enclosed manuscript has been revised in accordance with this feedback and detailed responses to each comment are provided below. A copy of the manuscript is provided with tracked changes as requested.

Comment 1: The manuscript is potentially interesting, but has major flaws in explanation of applied methods, and in the presentation and discussion of study results.

Response to Comment 1: Thank you for your interest in our paper. We have revised the manuscript according to your helpful suggestions.

Comment 2: Methods, Statistical Analysis: authors state that prevalence point estimates with 95% Confidence Intervals were obtained taking into account the complex sampling strategy. Presumably, authors accounted for sampling methods also when estimating prevalence differences (Table 5) or when applying logistic regression (text, Table 4): please specify.

Response to Comment 2: The text was revised to state that “All analyses were calculated using Stata v18 to account for complex sample survey design and to produce nationally representative estimates.” (Page 7).

Comment 3: As regards logistic regression, authors present aOR (adjusted odds ratio): adjusted for what (only in a footnote of Table 4 adjustment variables are reported)?

Response to Comment 3: The authors added the following sentence to the methods section: “Multivariable logistic regression was conducted to examine 1) factors associated with long COVID and 2) factors associated with prior diagnosis of COVID-19, adjusted for age, sex, race/ethnicity, education level, income, health insurance, geographic region, BMI, smoking status, disability, and depression.” (Page 7)

Comment 4: Results, Tables 2-5: please delete footnotes on smoking status, disability, depression (notes b-d in Table 2): the variables are extensively explained in the Methods, are notes are too lengthy

Response to Comment 4: The footnotes on smoking status, disability, depression have been removed from all tables for consistency, since they have been extensively explained in the Methods.

Comment 5: Results, Table 1: the most important study variable is missing, e.g. the prevalence of subjects ever diagnosed with COVID-19. The latter represents the denominator for the prevalence of long-COVID, and MUST be absolutely reported and discussed (including possible reasons for underdiagnoses and underreporting, and possible bias introduced in the study)

Response to Comment 5: The variable for prior COVID-19 diagnosis has been added to Table 1. Possible reasons for underdiagnoses and underreporting, and possible bias introduced in the study, have been added to the limitations section: “Second, COVID-19 vaccination, high-risk conditions, COVID-19 diagnosis, and long COVID symptoms were based on self-reports and were not verified by medical records, so may be subject to recall or social-desirability bias. Other nationally representative cross-sectional studies have shown that the reported prevalence of long COVID in 2022 was 6.9%-7.5%, which is lower than those found in this study. [32, 33] However, other studies demonstrate that the prevalence of long COVID can range from 7.5 to 41.0%, depending on how long COVID is measured and reported. [28] Over- or under-reporting of these variables may lead to misclassification, resulting in potential bias of the estimates.” (Page 13).

Comment 6: The same analyses reported in Table 1 should be therefore replicated for subjects reporting COVID-19 (see comment above)

Response to Comment 6: An additional column was added to Table 1 to demonstrate characteristics of adults with prior COVID-19 diagnosis.

Comment 7: Table 4 might be deleted: aOR are already reported in the text, and all the relevant information is already displayed in Table 3

Response to Comment 7: The authors would like to keep Table 4 in the manuscript because it provides adjusted odds ratios of the association between sociodemographic characteristics and each of the individual long COVID symptoms. In addition, a whole paragraph in the manuscript to dedicated to the results in Table 4. It is also important to keep Table 4 so that the reader can view other results that are not presented in the manuscript. (Pages 8-9)

Comment 8: Discussion is poor and must be more focused on study findings compared to previous evidence: as an example, authors cite studies reporting that long COVID is more likely with increasing age, which is the exact contrary of study results. Author might also cite a preprint based on BRFSS findings (doi 10.1101/2023.09.29.23296352).

Response to Comment 8: The discussion has been enhanced by providing additional comparisons to other studies, and citing further studies as suggested, including the preprint on BRFSS findings.

“This study found that long COVID was most common among younger age groups, females, NH American Indian or Alaska Native and NH other or multi-racial groups, those with lower annual household income, those with no health insurance, and those living in the Midwest or South, which is consistent with other studies. [19, 20]” (Page 11).

We also describe the possible differences between our results and those from other studies: “There were also differences by sociodemographic characteristics in common symptoms such as tiredness or fatigue, difficulty breathing or shortness of breath, loss of taste and smell, difficulty thinking or concentrating, or joint or muscle pain. Previous studies have found sex differences in long COVID, which could be related to differences in immune system function or sex hormones.[23, 24] Other studies found that long COVID was more likely with increasing age and BMI, which may also be related to immune function. [25, 26] While our study has found higher prevalence of long COVID among younger age groups which are consistent with results from other nationally representative surveys, [20] multiple factors could result in differences found in these studies, such as reporting of long COVID (self-report vs. doctor diagnosis), sample population (nationally representative data vs. medical records), and other sources of error (bias or misclassification).” (Page 11).

We also added additional sentences to the discussion to describe how these results can be applied in practice:

“Nevertheless, findings from this study offer several insights for practice. First, findings can aid in the identification of priority groups for intervention efforts. To our knowledge, prior health communication campaigns have not emphasized the risk of long COVID among groups that may be particularly vulnerable. We also note the need to provide guidance about individual rights, protections, and support for those experiencing long-term adverse effects from COVID-19. Promoting awareness of clinical guidance from the Department of Health and Human Services regarding treatment of those at greatest risk for developing long-COVID is an important part of a comprehensive approach. It has been well established that early COVID-19 testing and treatment can help prevent severe or long-term COVID-19 symptoms.

This study adds to the literature on the prevalence of long-term adverse health effects of COVID-19   and provides new information about population subgroups that may be at greatest risk. These findings corroborate calls to enhance efforts to diagnosis, treat, and mitigate the potential for adverse COVID-19 outcomes. Additionally, findings suggest the need to better educate the public about the importance of early treatment, the potential harms of COVID-19 infections, the benefits of vaccination, and characteristics of those at heighted risk to better inform health care service needs planning and other support services for those experiencing adverse side effects from long COVID.” (Pages 14-15).

Comment 9: In the second sentence of the introduction, authors cite an increase in hospitalization in December 2023, which was a too early finding at the time of the submission (the reference website was accessed in October 2023)

Response to Comment 9: The early date has been revised in the references.

Comment 10: The manuscript must be extensively edited, as an example, last sentence of the Introduction, “The identification OF groups…”, and other typos.

Response to Comment 10: This has been revised in the introduction: “The identification of groups with a higher prevalence of long COVID” (Page 4). The authors have reviewed the entire manuscript again to revise for other typos.

Reviewer 2 Report

Comments and Suggestions for Authors

I read the study with great interest and find it methodologically well-constructed. I was reflecting, more than anything, on the fact that in clinical practice, we classify long Covid as a diagnosis of exclusion, meaning it is necessary to eliminate other potential medical conditions. Since we lack a definitive method to diagnose long Covid, the approach involves ruling out all other possible causes for the patient's symptoms. Ultimately, when other conditions have been excluded, what remains is likely long Covid.

If, therefore, we don't have better approaches, when studying the prevalence of long Covid through surveys, I think there are limitations in both the subjectivity of the diagnosis (linked to the individual making the diagnosis, which in this case is reported by the patient themselves) and in the conditional presentation of the subject. One of the limitations of the long Covid diagnosis, in fact, is to investigate and eliminate other potential conditions that the patient may have (for example, experiencing fatigue even before, but perhaps attributing it to work, now to COVID). This is already a complex aspect for a doctor, let alone from a questionnaire. I would speak more about the prevalence of symptoms that align with those reported in the literature for long Covid than about the prevalence of long Covid itself. Or something similar. And I would adjust the entire paper accordingly, specifying this significant limitation, which is really lacking in the dedicated section. In general, all limitations should be better specified.

Another minor point is to understand that if the authors use the presence of a single dose to indicate the vaccination status, I believe it is partial. For two reasons. The vaccination schedule from the beginning (except for one vaccine) involved two doses; furthermore, we know that the vaccine tends to lose its effectiveness over time, and if that single dose was administered too long before, the effect is comparable to zero, especially if the questions were asked in 2022. In essence, considering individuals as vaccinated with "one or more doses" risks underestimating the effect of vaccinations, leading to a higher prevalence of symptoms "compatible" with long Covid in the vaccinated cohort.

Author Response

The authors greatly appreciate the prompt and constructive comments received by the reviewers. The enclosed manuscript has been revised in accordance with this feedback and detailed responses to each comment are provided below. A copy of the manuscript is provided with tracked changes as requested.

Comment 1: I read the study with great interest and find it methodologically well-constructed. I was reflecting, more than anything, on the fact that in clinical practice, we classify long Covid as a diagnosis of exclusion, meaning it is necessary to eliminate other potential medical conditions. Since we lack a definitive method to diagnose long Covid, the approach involves ruling out all other possible causes for the patient's symptoms. Ultimately, when other conditions have been excluded, what remains is likely long Covid.

If, therefore, we don't have better approaches, when studying the prevalence of long Covid through surveys, I think there are limitations in both the subjectivity of the diagnosis (linked to the individual making the diagnosis, which in this case is reported by the patient themselves) and in the conditional presentation of the subject. One of the limitations of the long Covid diagnosis, in fact, is to investigate and eliminate other potential conditions that the patient may have (for example, experiencing fatigue even before, but perhaps attributing it to work, now to COVID). This is already a complex aspect for a doctor, let alone from a questionnaire. I would speak more about the prevalence of symptoms that align with those reported in the literature for long Covid than about the prevalence of long Covid itself. Or something similar. And I would adjust the entire paper accordingly, specifying this significant limitation, which is really lacking in the dedicated section. In general, all limitations should be better specified.

Response to Comment 1: Survey data can provide useful information, particularly for those who may not have access to care. Many studies have collected self-reported prevalence of long COVID collected from survey questionnaires, including the Household Pulse Survey and the National Health Interview Survey. However, in line with the studies, the authors have added additional sentences in the limitations section to describe the limitations of using self-reported long COVID data: “Furthermore, the survey did not collect information on duration of symptoms, time since COVID-19 illness, or treatment during acute COVID infection, each of which could influence the reported prevalence of long COVID. Other nationally representative cross-sectional studies have shown that the reported prevalence of long COVID in 2022 was 6.9%-7.5%, which is lower than those found in this study. [30, 31] However, other studies demonstrate that the prevalence of long COVID can range from 7.5 to 41.0%, depending on how long COVID is measured and reported. [28] Over- or under-porting of these variables may lead to misclassification, resulting in bias of the estimates. Despite these limitations, these results provide invaluable data on experiences of those with long COVID and factors associated with long COVID, including among persons who might not have accessed care, and may complement studies based on administrative data.” (Page 12).

Comment 2: Another minor point is to understand that if the authors use the presence of a single dose to indicate the vaccination status, I believe it is partial. For two reasons. The vaccination schedule from the beginning (except for one vaccine) involved two doses; furthermore, we know that the vaccine tends to lose its effectiveness over time, and if that single dose was administered too long before, the effect is comparable to zero, especially if the questions were asked in 2022. In essence, considering individuals as vaccinated with "one or more doses" risks underestimating the effect of vaccinations, leading to a higher prevalence of symptoms "compatible" with long Covid in the vaccinated cohort.

Response to Comment 2: The questionnaire did not specify the brand of COVID-19 vaccines, or the date of administration of each vaccine. As a result, the effect of vaccination may be under-estimated, which has been added in the limitations section: “Fourth, the questionnaire did not specify the brand of COVID-19 vaccines, or the date of administration of each vaccine, in order to determine whether they have been up-to-date with all eligible doses. As a result, the effect of vaccination in this study may be under-estimated.” (Page 12).

Round 2

Reviewer 1 Report

Comments and Suggestions for Authors

The authors have addresses my major concerns. In the abstract, please rearrange the sentence "reporting having been diagnosed with long COVID": subjects were not diagnosed with long COVID, simply reported symptoms suggestive of / consistent with long COVID

Comments on the Quality of English Language

Minor editing is required, e.g.

1) Introduction, first paragraph, last sentence: "Long COVID can have major impacts ...." instead of "The impact of long COVID can have major impacts ...."

2) Introduction, towards the end of the second paragraph: please remove "status" from "body mass index (BMI) status"

3) Methods, Statistical Analysis, 3rd sentence: please remove "were assessed" from the end of the sentence

Author Response

Comment 1: The authors have addresses my major concerns. In the abstract, please rearrange the sentence "reporting having been diagnosed with long COVID": subjects were not diagnosed with long COVID, simply reported symptoms suggestive of / consistent with long COVID

Response to Comment 1: This sentence has been revised as suggested: “Overall, more than one in five adults who ever had COVID-19 reported symptoms consistent with long COVID (21.8%).” (Page 1)

Comment 2: 1) Introduction, first paragraph, last sentence: "Long COVID can have major impacts ...." instead of "The impact of long COVID can have major impacts ...."

Response to Comment 2: This sentence has been revised as suggested: “Long COVID can have major impacts on mental health, quality of life, and the ability to work.” (Page 1)

Comment 3: 2) Introduction, towards the end of the second paragraph: please remove "status" from "body mass index (BMI) status"

Response to Comment 3: This has been revised as suggested: “it is unknown whether depression, smoking status, body mass index (BMI), and other factors” (Page 2)

Comment 4: 3) Methods, Statistical Analysis, 3rd sentence: please remove "were assessed" from the end of the sentence

Response to Comment 4: This has been revised as suggested: “The difference in proportion of adults experiencing each outcome was compared across levels of each variable (e.g., the proportion with long COVID among adults aged 18–49 years compared to 65 years and older).” (Page 3).

Reviewer 2 Report

Comments and Suggestions for Authors

Authors fully addressed the my comments.

Author Response

Comment: Authors fully addressed the my comments.

Response: We thank you for your review.